# Would Climate Change Influence the Potential Distribution and Ecological Niche of Bluetongue Virus and Its Main Vector in Peru?

**DOI:** 10.3390/v15040892

**Published:** 2023-03-30

**Authors:** Dennis A. Navarro Mamani, Heydi Ramos Huere, Renzo Vera Buendia, Miguel Rojas, Wilfredo Arque Chunga, Edgar Valdez Gutierrez, Walter Vergara Abarca, Hermelinda Rivera Gerónimo, Mariano Altamiranda-Saavedra

**Affiliations:** 1Laboratorio de Microbiología y Parasitología—Sección Virología, Facultad de Medicina Veterinaria, Universidad Nacional Mayor de San Marcos, Lima 15001, Peru; 2Laboratorio de Inmunología, Facultad de Medicina Veterinaria, Universidad Nacional Mayor de San Marcos, Lima 15001, Peru; 3Laboratorio de Referencia Nacional de Metaxenicas y Zoonosis Bacterianas, Centro Nacional de Salud Pública, Instituto Nacional de Salud, Lima 15001, Peru; 4Laboratorio de Sanidad Animal “M.V. Atilio Pacheco Pacheco”, Escuela Profesional de Zootecnia, Universidad Nacional San Antonio Abad del Cusco, Cusco 08681, Peru; 5Grupo de Investigación Bioforense, Tecnológico de Antioquia Institución Universitaria, Medellín 050005, Colombia

**Keywords:** Peru, bluetongue virus, *Culicoides insignis*, ecological niche, climate change

## Abstract

Bluetongue virus (BTV) is an arbovirus that is transmitted between domestic and wild ruminants by *Culicoides* spp. Its worldwide distribution depends on competent vectors and suitable environmental ecosystems that are becoming affected by climate change. Therefore, we evaluated whether climate change would influence the potential distribution and ecological niche of BTV and *Culicoides insignis* in Peru. Here, we analyzed BTV (*n* = 145) and *C. insignis* (*n* = 22) occurrence records under two shared socioeconomic pathway scenarios (SSP126 and SSP585) with five primary general circulation models (GCMs) using the kuenm R package v.1.1.9. Then, we obtained binary presence–absence maps and represented the risk of transmission of BTV and niche overlapping. The niche model approach showed that north and east Peru presented suitability in the current climate scenario and they would have a decreased risk of BTV, whilst its vector would be stable and expand with high agreement for the five GCMs. In addition, its niche overlap showed that the two niches almost overlap at present and would completely overlap with one another in future climate scenarios. These findings might be used to determine the areas of highest priority for entomological and virological investigations and surveillance in order to control and prevent bluetongue infections in Peru.

## 1. Introduction

Bluetongue (BT) is a vector-borne disease caused by bluetongue virus (BTV) and is transmitted by *Culicoides* spp. (Diptera; Ceratopogonidae) to ruminants [1]. BTV belongs to the *Orbivirus* genus within the *Reoviridae* family and is a non-enveloped virus with 10 segments of double-stranded RNA (dsRNA) [1]. Despite there being twenty-seven identified serotypes, eight additional potential serotypes have been suggested [1,2,3]. BTV can affect several domestic (cattle, sheep and goats) [4,5] and wild ruminants (bighorn sheep, deer and pronghorn antelope), as well as some camelids (alpaca) [6,7]. In general, BT severe illness is mainly restricted to certain breeds of sheep and white-tailed deer; so, they can manifest many of the signs including fever, nasal discharge, hyperemia and edema of the lips, ears, face and submaxillary region, erosions of the oral mucosa, reduced milk yield and abortions [1,8,9]. Cattle play an important role in BTV epidemiology because of prolonged and persistent viremia from 7 to 63 days without clinical signs; so, they are considered to be reservoirs [5]. In addition, cattle enhance the spread of BTV as *Culicoides* spp. preferentially feed on them [9,10].

BTV is listed in the notifiable terrestrial and aquatic animal diseases of the World Organization for Animal Health (WOAH founded OIE) [11]. It has a negative economic impact on countries and also on the health of threatened wild populations [12]. BTV affects the national and international trade of animals; for instance, loss costs amounted to EUR 40 million during a 2007 outbreak in Germany with fatality rates up to 13.1% for cattle and 41.5% for sheep [13,14]. Regarding South America, BTV has been reported in Ecuador, Argentina, Colombia, French Guiana, Brazil and Peru [15]. In Brazil, BTV outbreaks have been reported since 2001 in sheep and goats that showed clinical signs, where some of them died [16]; furthermore, BTV caused significant mortality (18.4%) in 2014 [17,18]. In addition, dwarf brocket deer (*Mazama nana*) were affected by hemorrhagic disease during a BTV outbreak in 2015–2016 [19]. Conversely, Peru has not reported any clinical evidence of BT, although infection and seropositive animals have been documented in tropical regions [20,21].

The worldwide distribution of BTV depends on competent *Culicoides* vectors and suitable environmental ecosystems [22]. *Culicoides sonorensis* (With and Jones 1957) is the main vector of BTV in North America [23], while *Culicoides insignis* (Lutz 1913) is the main form in the southeastern United States of America, South America (SA) and the Caribbean [23,24]; *Culicoides pusillus* (Lutz 1913) could also be involved with BTV in SA [25,26]. Biting midges are blood-sucking insects that are often associated with farm environments close to muddy areas or along the margins of vegetated ponds among other areas [23,26,27]. Regarding *C. insignis*, it is significantly more abundant in livestock farms compared with both poultry and sylvatic habitats [28]. In addition, temperatures between 20 and 25 °C are associated with an abundance that refers to spring and summer seasons, and variables such as precipitation, wind speed and rainfall affected its highest abundances [29]. These variables, as well as agricultural and animal husbandry patterns, have been related to the presentation of BT infection [22,23].

Understanding the factors that are related to the geographic distribution of diseases is possible with ecological niche modeling (ENM) and species distribution modeling (SDM) [30,31,32]. Basically, SDM is the process of estimating the actual or potential distribution area, or set of suitable habitats for a species, based on its observed presences and (sometimes) absences [33]. In addition, ENM can estimate the Grinnellian fundamental niche that represents the environmental conditions needed for long-term population persistence [30,33]. However, certain assumptions should be considered, such as biotic interactions that could be neglected which denote “the Eltonian noise hypothesis” under the biotic, abiotic and movement (BAM) framework [33,34]. In addition, forecasting distributions and distributional changes in diverse spatial scales is possible with ENM approaches, but such forecasts should come from analyses of extensive data [30].

ENM and SDM have already been used to predict the potential distribution of BTV in the current [31,35] and future climate scenarios [31]. SDMs have also been generated for some *Culicoides* spp., such as *Culicoides imicola* (Kieffer 1913), *Culicoides varipennis* (Wirth and Williams 1957), *C*. *sonorensis*, *C. insignis* and others [31,36,37,38], highlighting abiotic (precipitation, temperature and seasonality) and biotic factors (sheep distribution or livestock density) as being the most important predictors for BTV and *Culicoides* spp. [31,35,36,37]. Therefore, it would be possible to determine the areas of the highest priority for the control and prevention of BTV in Peru using these tools, hence reducing personnel and equipment costs. These analysis approaches provide an opportunity to identify the factors that affect the spatial and temporal distribution of the vectors and hosts of BTV, thus promoting knowledge of the epidemiology and dynamics of virus transmission to domestic and wild ruminants. For this reason, our main aim was to evaluate whether climate change would influence the potential distribution and ecological niche of BTV and *C. insignis* in Peru.

## 2. Materials and Methods

### 2.1. Occurrence Records

BTV occurrence records (*n* = 513 individual animals) were obtained from ruminants’ serum (cattle, goats and sheep) via field sampling using competitive ELISA for antibody detection from 24 departments of Peru via Servicio Nacional de Sanidad Agraria (SENASA) from June 2017 to October 2019, and the virology laboratory at Universidad Nacional Mayor de San Marcos (UNMSM) was also used for occurrences (*n* = 52) from July 2021 to July 2022. *Culicoides insignis* collections were performed in different farms of Peru using mini-CDC light traps from July 2021 to July 2022, with 15 occurrences registered. We also acquired one *C. insignis* record from the scientific literature [39] and 13 from speciesLink (https://specieslink.net/, accessed on 16 September 2022). The geographic coordinates of each location were obtained using a GPS unit; however, for obtaining records from the database, we assigned them based on the consultation of online gazetteer data (www.gpsvisualizer.com (accessed on 16 September 2022)).

### 2.2. Occurrence Records Cleaning

BTV occurrence records were filtered to eliminate duplicates or georeferencing errors, and we also reduced the data to avoid problems with spatial autocorrelation; so, we rarefied based on 20 km buffer that was proposed by Samy and Peterson [31] using the spThin R package [40]. After these data cleaning steps, we had 145 and 13 occurrence records from 2017 to 2019 and 2021 to 2022 data, respectively (Appendix A). Then, we randomly split the 145 records into two sets for model calibration (80%) and evaluation (20%). The 13 occurrences represented BTV independent data for model validation (Appendix A). Similarly, *C. insignis* occurrence records (*n* = 29) were filtered and rarefied based on a 2 km distance that represented the average flight distance for *Culicoides* [41]. At the end, we had data for *C. insignis* from 22 occurrences (Appendix A).

### 2.3. Calibration Area

The accessible area (M) was delimited based on our occurrence data. Therefore, we created the virus’ calibration area taking the elevation at which seropositive animals for BTV were found; hence, we used a DEM raster from Worldclim at a 30 arc-second and maximum elevation of 2888 m above sea level. For *C. insignis*, however, we relied on a map of the terrestrial ecoregions of the world [42]; therefore, each occurrence record was overlaid on the ecoregion map to shape its “M” area.

### 2.4. Environmental Data

Bioclimatic variables were gathered from CHELSEA V.2.1 [43] (https://chelsa-climate.org/ (accessed on 16 September 2022)) in the form of 19 bioclimatic data layers at a 30 arc-second (~1 km) of spatial resolution. Both current (1981–2010) and future climate scenarios (2071–2100) were downloaded under two shared socioeconomic pathway scenarios (SSP126 and SSP585), to represent an optimistic scenario that assumed climate protection measures and the dramatic scenario of the range of pathways with fossil-fueled development, respectively [44]. In addition, five primary general circulation models (GCMs) were used in our prediction model (GFDL-ESM4, IPSL-CM6A-LR, MPI-ESM1-2-HR, MRI-ESM2-0 and UKESM1-0-LL), as recommended by the literature [43].

We used four and five sets of bioclimatic variables for BTV and *C. insignis*, respectively (Appendix A). Following these criteria, we obtained (a) Set 1: 19 variables from CHELSA v.2.1; (b) Set 2: jackknife processes in Maxent to select variables that contributed most to models (>80%); (c) Set 3: pairwise Pearson’s correlation coefficients (r), thereby eliminating one variable per pair with the value of |r| being less than 0.80; (d) Set 4: variables with VIFs (variance inflation factors) less than 10 [45] and (e) Set 5 (only for *C. insignis*): variables used for the construction of ENMs of other species of medical importance such as sandflies and culicids [46].

### 2.5. Ecological Niche Modeling and Model Transfers

The maximum entropy (Maxent v.3.4.4) algorithm was used to model an ecological niche using the kuenm R package v.1.1.9 [40]. We combined four and five sets of environmental variables for BTV and *C. insignis*, respectively, 17 values of regularization multiplier (0.1–1 with intervals of 0.1; 2–6 with intervals of 1, 8 and 10) and 7 possible combinations of 3 feature classes (linear = l, quadratic = q and product = p) [40]. The candidate model performance and best model selection were based on the significance of the partial receiver operating characteristic (pROC), with 500 iterations and 50% of the data for bootstrapping, prediction ability (omission rate, E = 5%), model fit and complexity AICc (Akaike’s information criteria), delta AICCc and weight AICCc [40].

The final model creation and its evaluation for each species was obtained using 10 replicates via bootstrap with raw outputs; these were transferred from the accessible area “M” to the projection area “G”—Peru. Then, the type of model output for BTV and *C. insignis* (free extrapolation, extrapolation and clamping, and no extrapolation) was selected based on the two important biological variables (Bio 1: mean annual air temperature and Bio 12: annual precipitation amount); so, they were plotted in a two-dimensional space, along with the occurrences, to determine their position in the environmental space (Appendix A) [47]. After that, the median of the GCMs (GFDL-ESM4, IPSL-CM6A-LR MPI-ESM1-2-HR, MRI-ESM2-0 and UKESM1-0-LL) was used to reclassify the maps into binaries for generating a consensus of each future climate scenario (SSP126 and SSP585). It was based on a threshold that omitted all regions with habitat suitability below the suitability value, using the 5 % omission error rate threshold (based on the lowest training presence threshold approach), assuming that this value refers to the percentage of data that may include errors misrepresenting the species’ environment [48].

Furthermore, the identification of changes in suitable areas and suitability in projections was calculated and represented by agreement maps of changes (stable, gain and loss); therefore, we compared five GCMs against the current projection and quantified the agreement of the gain and loss of suitable areas, as well as the stability of suitable and un-suitable conditions using the function “kuenm_projchanges” of the kuenm R package v.1.1.9 [40]. Next, the degree of extrapolation (i.e., prediction areas where environmental conditions differ from those represented within the training data [49]) was computed via a multivariate environmental similarity surface (MESS) using the kuenm R package v.1.1.9 [40]. We binarized with 10% of threshold using the average map of each SSP-GCM replicate. These were used to cut their respective maps of BTV or *C. insignis* in current and future climate scenarios (SSP126 and SSP585). In addition, the final model for BTV in the projection area was evaluated after being created; for this step, independent data from samples of 2021–2022 were used.

### 2.6. Ecological Niche Overlap

The environmental spaces were visualized using the software Niche Analyst (NicheA) version 3.0 [50], available at http://nichea.sourceforge.net/ (accessed on 20 October 2022). A three-dimensional ellipsoid of environmental distribution was created for BTV and *C. insignis* in a virtual space of bio-climatic conditions, represented by three principal components (PCAs). The PCAs of the 19 bioclimatic variables from CHELSA V.2.1 were represented by PC1, PC2 and PC3, and we quantified similarity between the niches in terms of overlap using NicheA, whilst also estimating the Jaccard index for each one [51].

### 2.7. Risk Map Design

We used binary maps of the raw outputs of the best final models cut with MESS and then we classified the data into three categories: (1) low risk, which refers to areas not suitable for either BTV or *C. insignis*; (2) moderate risk, which means suitable areas only for BTV or *C. insignis*; and (3) high risk, which refers to suitable areas for both. In addition, the percent of the number of grids was calculated for each category and represented in a pie chart.

## 3. Results

### 3.1. Model Calibration Results

We obtained 476 and 595 candidate models for BTV and *C. insignis* in the calibration area. Then, three and six models were found as the final model candidates for BTV and *C. insignis*, respectively. Afterward, we selected the best model for each one based on the lowest ∆AICc. Therefore, the models that worked better with the variables in Set 4 were selected for both species (Table 1). The final model validation for BTV in the projection area of the current climate scenario showed a pROC equal to 0 and a mean AUC ratio of 1.32.

### 3.2. Potential Suitable Areas for Bluetongue Virus and C. insignis under Current and Two Future Climate Scenarios (SSP126 and SSP585)

The current distribution area of BTV in Peru is mostly concentrated in the east and north of the country. The BTV location mainly covers eleven of twenty-four political divisions and the ecoregions that presented suitability for the occurrences were Ucayali moist forests, Eastern Cordillera Real montane forests, Southwest Amazon moist forests, Marañón dry forests, Peruvian Yungas, Sechura Desert, Napo moist forests, Central Andean wet puna and Tumbes–Piura dry forests. In addition, the suitable area in Peru accounts for 77.42% in the current climate scenario (Figure 1A), and it would be decreased to 31.42% (Figure 1B) or 12.57% (Figure 1C) depending on whether the future climate scenarios were SSP126 or SSP585 by 2071–2100, respectively.

Regarding *C. insignis*, its current distribution mostly covers eastern Peru and eight of the twenty-four political divisions. In the case of the ecoregions, suitability is possibly found in Ucayali moist forests, Eastern Cordillera Real montane forests, Southwest Amazon moist forests, Peruvian Yungas and Marañón dry forests (Figure 2A). Conversely to BTV, the suitable area for *C. insignis* in the current climate scenario would be increased from 66.33% to 87.02% in the SSP126 future climate scenario, or to 75.80% in the SSP585 by 2071–2100.

Additionally, future model transfers showed more stable suitable areas for *C. insignis* (i.e., suitable in current and future climate scenarios) than BTV across the Peruvian east (Figure 3 and Figure 4). The areas of range reduction (loss) were higher in BTV and mainly on the coast and the eastern zone (Figure 3A,B), whereas there was no evidence of a reduction in SSP126 for *C. insignis* (Figure 4A), but SSP585 showed few reductions in the center of Peru (Figure 4B). A range expansion (gain) was evidenced for *C. insignis* with high agreement for the five GCMs; however, BTV distribution showed that the gain of suitable areas was concentrated in the northern parts of Peru (Figure 3A,B). In general, we noted greater agreement among the five GCMs in terms of losses for BTV and gains for *C. insignis* in both SSP126 and SSP585 scenarios.

### 3.3. Uncertainty in Current and Future Climate Scenarios

MESS analysis detected environmental areas that were similar and different between model training and model projection data. For BTV, different environmental areas were shown in the middle that represented the Andean highlands of Peru (Figure 1A); however, future climate scenarios (SSP126 and SSP585) showed areas of no extrapolation in the coast and some parts of the eastern area (Figure 1B,C). Regarding *C. insignis*, MESS results exhibited high agreement in non-extrapolative areas among future scenarios, with most of these areas concentrated on Peru’s coast. In addition, sources of variability results indicated no variation coming from GCMs, SSPs or replicates (Appendix A), with the only sources of variation documented for *C. insignis* coming from replicates (Appendix A).

### 3.4. Risk Map in Current and Future Climate Scenarios

The BTV risk map in Peru showed that high risk areas are mainly in the east of the country (current climate scenario) and represent the highest coverage of Peruvian territory (Figure 5A). However, the high risk would decrease gradually from current to SSP126 and SPP585 future climate scenarios, except for the northeastern corner of Peru (Figure 5B,C), and it would increase the moderate risk by 2071–2100 in both scenarios. All of the scenarios would show low-risk areas that are lower than the others, as represented in the top of each map as a pie chart (Figure 5).

### 3.5. Ecological Niche Overlap

Minimum-volume ellipsoid (MVE) niche estimations showed that the two niches almost overlap in the current climate scenario (Figure 6A) and would completely overlap with one another in future climate scenarios (Figure 6B,C). The MVE reconstructions demonstrate that the niche would undergo an important shift in the SSP585 scenery. Additionally, the degree of overlap using the Jaccard similarity index was generally low. Values were 0.29 and 0.21 in the scenarios SSP585 and SSP126, respectively, and 0.12 in the current climate scenario. Regarding niche amplitudes, BTV had a higher niche compared to *C. insignis* in all of the scenarios (Figure 6).

## 4. Discussion

The species distribution modeling for BTV and *C. insignis* in the current climate scenario allowed potential areas that mainly cover the east of the country to be identified. Peru is located in the Neotropic and its eastern region is part of the Amazon basin, with the main biome being tropical and subtropical moist broadleaf forest [42], where BTV infection is considered to be endemic because of the presence of its *Culicoides* vector [1]. This area has a temperature that promotes faster rates of virogenesis with more rapid BTV replication [52,53] and it would also decrease the intervals between feedings, thereby increasing the chances for potential transmission. In addition, egg and larval development of *Culicoides* spp will be faster and the higher temperatures may shorten the lifespan of the female [23,54,55]. The BTV potential current distribution was also described by Samy and Peterson [31] but the focus was worldwide. Their results were different from our results as they showed that BTV would be present along the Andes region of Peru (center), possibly because they did not use any occurrences from Peru with most of them being from Europe and the United States. Regarding *C. insignis*, its potential current distribution was evaluated in previous studies that were modeled for northwestern Argentina [56] and Florida [36]. Therefore, this study represents the first potential distribution of *C. insignis* in the northwest of South America.

Furthermore, the suitable areas for BTV and *C. insignis* were mainly represented by the ecoregions of Ucayali moist forests and Southwest Amazon moist forests. These ecoregions are part of the rainforest of Peru where the altitude is above 500 masl and it has a tropical climate, with high temperatures and rainfall throughout the year [57]. The average annual rainfall is between 1000 and 3000 mm, with temperatures ranging from 22 °C in the eastern Andes to 31 °C in the Amazon [58]. The Peruvian Amazon is home to numerous *Culicoides* species, which prefer moist regions, grass and decomposing material such as the remains of banana trees [59]. This part of the planet showed 64% more deforestation in 2020 than in 2019 as registered by Brazilian satellites [60]. This increases the likelihood of environmental degradation, heightens population health risks and causes ecosystem loss on a local, regional and international level. Therefore, these suitable areas should be monitored by SENASA to establish passive surveillance programs for BTV and its vectors.

On the one hand, the suitable area for *C. insignis* would increase in both future climate scenarios with high agreement among the five models. The *Culicoides’* response to climatic variables is different between midge species and midge species groups [61,62]. In addition, the emergence of BTV in Europe has been particularly attributed to climate change because of the range of expansion of *C. imicola* [23,63,64], and these findings could also be possible due the more intense sampling and the use of more effective traps because of the BTV outbreaks. However, there are no studies of *C. insignis* adaptation to changing climate, but there is evidence of records located outside of this historical range [36]. In our results, *C. insignis* would invade the central Andes, where it could find naive populations and cause a massive outbreak and the subsequent death of cattle, similar to that seen for *Culicoides obsoletus* (Meigen 1918) in the western Palearctic region in 2006–2008 [65]. In addition, Peru is one of the 20 nations that are most susceptible to climate change and its consequences can already be seen in many places [66]. For example, temperature records from 1981 to 2010 showed a warming trend of more than 0.2 °C per decade in the central Andes [67]. Furthermore, *C. insignis* is a versatile midge; hence, it may utilize more diverse larval development sites including mangrove swamps, sugarcane fields and a variety of streams, springs, ponds and ditches of both natural and human origin [23].

On the other hand, our results showed that a BTV-suitable area would decrease in both future climate scenarios and the MESS analysis cut the coastal zone of Peru despite the presence of occurrence records in the current climate scenario. The coastal zone is represented by the Sechura desert, Peruvian coastal desert and the Atacama desert, which are nearly devoid of rain and below 1000 masl, with a coastal topography that is low and flat [68]. These areas are environmentally different from those where BTV positive cases are frequently observed in Peru [20,21,69]. In addition, the occurrence records were obtained via serological analysis that does not have 100% sensitivity and specificity [70]. Additionally, the occurrence records from the coastal zone could come from migrations as a result of the movement of animals in the country.

The current risk map in Peru showed high overlap between BTV and *C. insignis* areas. These areas represent endemic spaces of the country for BTV where its presence has been documented at the serological and molecular level [20,21,69]. The high risk would decrease from current to future climate scenarios, except for the northeastern corner of Peru where it would be possible because a new competent vector could emerge similar to the case of *C*. *obsoletus* in the northern region of Europe [71]. The moderate risk would remain latent in the future climate conditions and it would cover a larger part of the country. Surveillance programs into *C. insignis* and the host have to be performed because BTV is a double-stranded RNA virus with 10 segments, so reassortment plays an important role in the overall genetic diversification of BTV [72]. In addition, previous studies have documented that reassortment in some *Culicoides* was reportedly higher (42% in *C*. *sonorensis*) than in vertebrate hosts (5% in sheep) [73,74,75]. Furthermore, we consider the importance of adding a density layer of cattle, goat and sheep to improve the prediction of BTV and *C. insignis* distribution, as this species is the most abundant and frequently associated with livestock facilities [10]. In addition, it is important to establish more exhaustive sampling and use more effective capture traps for an adequate estimate of the biodiversity of *Culicoides* associated with livestock.

In the environmental space, BTV and vector populations appear to share similar ecological niches, but BTV has a higher niche compared to *C. insignis* in all of the scenarios. This might be due to the potential presence of another competent vector. Currently, *C. insignis* is probably the only competent vector of BTV in South America [76], and it has been recorded predominantly in the eastern part of Peru [39,77]. Although *C*. *pusillus* has also been recorded in the country [77], its role as a competent vector of BTV remains unknown [22,26]. In addition, BTV transmission is not only restricted to being vector-borne, as it can also be transmitted by direct contact with the atypical BTV serotypes [78,79]. Moreover, BTV is usually transmitted for more than one species in one continent such as *C*. *imicola* and *C*. *obsoletus* in Europe [41,80], *Culicoides brevitarsis* (Kieffer 1917), *Culicoides fulvus* (Sen and Das Gupta 1959), *Culicoides wadai* (Kitaoka 1980), *C*. *obsoletus*, *C*. *imicola* and *Culicoides pulicaris* (Linnaeus 1758) in Asia [41,80] *C*. *sonorensis*, *Culicoides stellifer* (Coquillett 1901) and *C. insignis* in North America [23,41,76,81]. For these reasons, medical veterinary entomology has to be developed in Peru and should incorporate the ecological dimension into management and arbovirus/vector control strategies to determine the mechanisms involved in BTV transmission.

## 5. Conclusions

We presented ecological niche models for BTV and its main vector *C. insignis* in Peru. This modeling strategy enabled us to evaluate the effect of climate change in BTV and *C. insignis* distribution for the period 2071–2100, that would remain with a moderate risk (green) in much of Peru. These findings might be used to determine the areas of the highest priority for entomological and virological investigations and surveillance in order to control and prevent bluetongue infections in Peru.

## Figures and Tables

**Figure 1 viruses-15-00892-f001:**
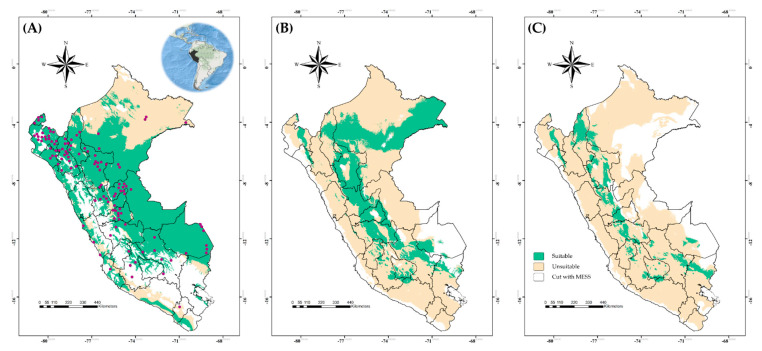
Suitable and unsuitable areas for the potential distribution of bluetongue virus in Peru under (**A**) the current climate scenario, and (**B**) the climate change in 2071–2100 in the future climate scenarios of SSP126 and (**C**) SSP585. Occurrence records (purple points).

**Figure 2 viruses-15-00892-f002:**
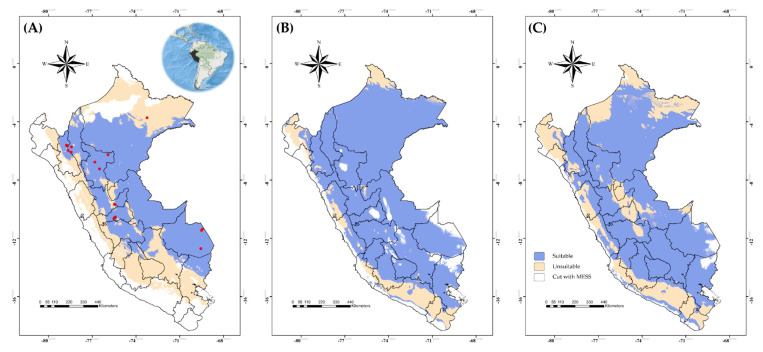
Suitable and unsuitable areas for the potential distribution of *Culicoides insignis* in Peru under (**A**) the current climate scenario, and (**B**) the climate change in 2071–2100 in the future climate scenarios of SSP126 and (**C**) SSP585. Occurrence records (red points).

**Figure 3 viruses-15-00892-f003:**
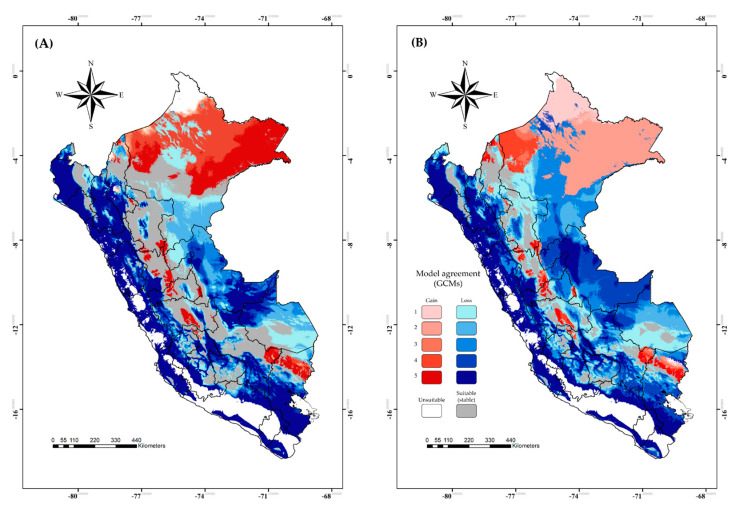
Comparison of binary models considering agreement among five GCMs for the potential geographic distribution of bluetongue virus based on the current and future climate in two distinct scenarios: (**A**) SSP126 and (**B**) SSP585. A blue scale means an agreement level of five GCMs which are no longer suitable, while a red scale denotes them being newly suitable. Gray color means suitable in the current climate scenario and for all GCMs, while white denotes being unsuitable in the current climate scenario and for all GCMs.

**Figure 4 viruses-15-00892-f004:**
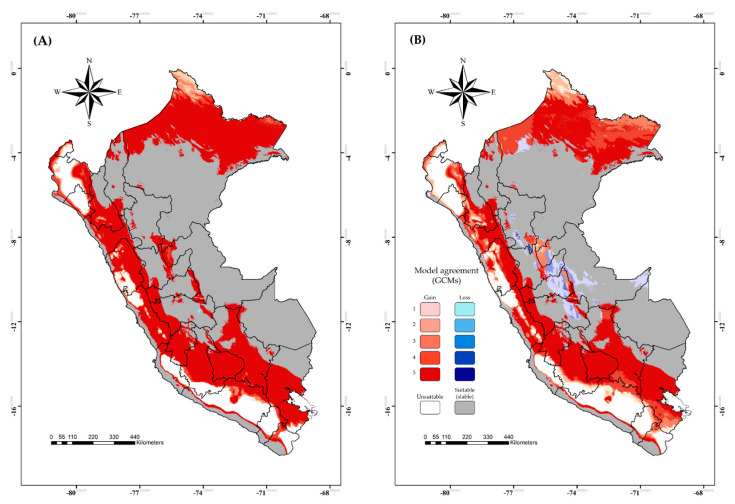
Comparison of binary models considering agreement between five GCMs for the potential geographic distribution of *Culicoides insignis* based on the current and future climate in two distinct scenarios: (**A**) SSP126 and (**B**) SSP585. A blue scale means an agreement level of five GCMs which are no longer suitable, while a red scale denotes them being newly suitable. Gray color means suitable in the current climate scenario and for all GCMs, while white denotes unsuitable in the current climate scenario and for all GCMs.

**Figure 5 viruses-15-00892-f005:**
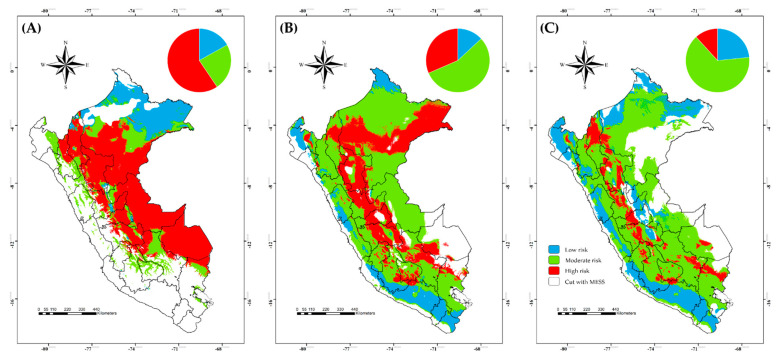
Risk map of bluetongue virus transmission by *Culicoides insignis* in Peru based on current (**A**) and future climate in two distinct scenarios (**B**) SSP126 and (**C**) SSP585. Colors represent different risk levels.

**Figure 6 viruses-15-00892-f006:**
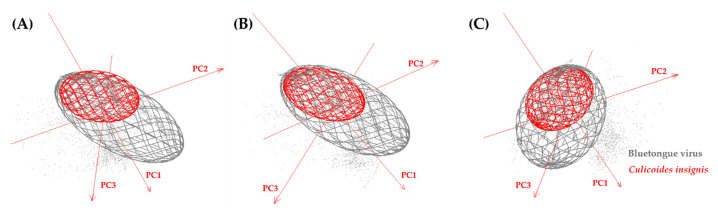
Minimum-volume ellipsoid (MVE) niche estimations for bluetongue virus and *Culicoides insignis* in three-dimensional environmental space in Peru based on the current (**A**) and future climate in two distinct scenarios: (**B**) SSP126 and (**C**) SSP585. Gray points represent environmental background conditions.

**Table 1 viruses-15-00892-t001:** Description of metrics of the selected models for each species.

Species	Occurrence Records	Model Settings	Set of Variables	*p*-ValuepROC	E%	ΔAICc	Parameters
**Bluetongue virus**	145	RM = 0.8; FC = lp	Set 4: Bio 2, Bio 3, Bio 4, Bio 8, Bio 15, Bio 18 and Bio 19	0	0.17	0	14
** *Culicoides insignis* **	22	RM = 0.8; FC = q	Set 4: Bio 2, Bio 3, Bio 4, Bio 8, Bio 13, Bio 15, Bio 18 and Bio 19	0	0.00	0	5

RM: regularization multiplier; FC: feature classes (linear = l, quadratic = q, product = p); pROC: partial receiver operating characteristic; E%: omission rate at 5%; ΔAICc: delta Akaike information criterion corrected for small sample sizes. Bio 2: mean diurnal air temperature range; Bio 3: isothermality; Bio 4: temperature seasonality; Bio 8: mean daily mean air temperatures of the wettest quarter; Bio 13: precipitation amount of the wettest month; Bio 15: precipitation seasonality; Bio 18: mean monthly precipitation amount of the warmest quarter; Bio 19: mean monthly precipitation amount of the coldest quarter.

## Data Availability

The data used for this study are included in the supplementary information files.

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
