# Peer review of "Would Climate Change Influence the Potential Distribution and Ecological Niche of Bluetongue Virus and Its Main Vector in Peru?"

_viruses, 2023, doi:10.3390/v15040892_

Round 1

Reviewer 1 Report

The manuscript discusses the potential influence of climate change on the ecological niche of bluetongue virus and Culicoides insignis in Peru. Information on the occurrence of bluetongue and Culicoides species are relative scarce, and this manuscript will help to fill this gap. The manuscript is well written and warrants publication.

Some minor suggestions may be:

It may depend on the discretion of the journal, but bluetongue (and other virus and diseases names) is usually not written with capital letters. The exception being viruses and diseases named after a person or geographical area (e.g., Schmallenberg virus or African horse sickness). This is similar to human diseases e.g., like measles or influenza.

The genus “Culicoides” must be given in full the first time a new species is mentioned. Similarly, the author’s name of the species needs to be given the first time is mentioned. E.g., Culicoides insignis Lutz. Check throughout manuscript. Ensure that Culicoides names (genus and species) are given in italics throughout the manuscript.

Line 37 (Diptera; Ceratopogonidae) can be inserted after spp.

Line 58: “affected by hemorrhagic disease” Do you refer to Epizootic hemorrhagic disease (EHD) or some other hemorrhagic disease? Other Culicoides transmitted disease e.g., Oropouche virus (OROV) and Epizootic hemorrhagic disease can be mentioned briefly. E.g., low/high abundance, absent or very restricted distribution area.

Line 85: “Bluetongue virus” can be given as “BTV”.

Line 104 – 110: Was all these collections made with the same model of light trap? or were other collections methods also utilised? Also state if all the collections were made near livestock.

Line 120: “biting midges” can be deleted.

Line 305-307: It can perhaps be added that the faster rates of virogenesis will also decrease the intervals between feedings thereby increasing the chances for potential transmission. It can be added that higher temperatures may shorten the lifespan of the female.

Line 326: “potential” can be inserted before “vectors”.

Line 329 – 331: The outbreaks of southern Europe was INITIALLY ascribed to the northward expansion of C. imicola, considered as the only competent vector of BTV at that time. Outbreaks in central Europe, in the absence of C. imicola, has, however, shown that several European Culicoides species can transmit bluetongue virus (because of climate change?). The finding of C. imicola in areas where it was not collected before, could have been the result of more intense sampling (due to the outbreaks) and the use of more effective traps.

Line 366: The apparent absence of C. insignis in BTV infected areas can potentially also be the result of under sampling. More effective light traps used on a regular basis near livestock may indicate higher species diversity.

See: McDermott, E.G. & Lysyk, T.J. (2020) Sampling considerations for adult and immature Culicoides (Diptera: Ceratopogonidae). Journal of Insect Science, 20, 1–11.

Line 369: It can once again be mentioned that it is also one of the most abundant livestock species in Peru.

Line 416” Check the spelling of “Maclachlan”. It must be “MacLachlan”.

Line 468: “Purse, B. v.” must be “Purse, B.V.,”

Line 516: Check the use of capital letters in the title of the reference.

Author Response

Dear, Reviewer 1

Thanks for your suggestions, I attached my response

Reviewer 2 Report

Authors used various statistically modeling, to predict and construct the risk map of the current situations versus the future optimistic and dramatic scenarios of BTV infection in Peru.  In Figure 5, they show that the risk of BTV transmission, by C. insignis (assuming that no new competent vectors will emerge), is shrinking, except for that  the north eastern corner of Peru (Figure 5B) will newly become the high risk area, if the modeling is correct.

Most of the materials and methods and results are not related to virology or BT the diseases.  The introduction and discussion are well written.

Section 2.1: the so-called "occurrence records" using ruminant serum.  Please clarify 1. the n = 513 means sera from 513 individual animals or each occurrence have variable number of animal serum? 2. you tested serum for antibody using what methods or you used molecular detection, or both?  Why not use "infection"? or seroprevalence rate or molecular detection rate?  Since Peru have not had any reported outbreak of BT (line 59-60), that means the breeds of sheep population is not sensitive to BTV, I think it is better for you describe the breeds of these sheep in Peru, unless in the future other sensitive sheep breeds are introduced and will of course influence the potential outcome.

Section 2.2: after record cleansing, are 145 and 13 appropriate for such extensive modeling.  What I mean is these numbers statistically valid?

line 114: how was the 20 km set? what reasons did you use.

line 119: how was the 2 km set? what reasons did you use.

line 358: since you mention that BTV has 10 segments dsRNA that tend to reassort, I am wondering if the tools you used to analyze in Figures 1 and 3, will be influenced by whether they are DNA or RNA virus, single linear vs segmented, single strand vs double strand, arthropod borne vs non-arthropod borne?  Will these affect the ultimate results of modeling presented in Figure 5?

Author Response

Dear, Reviewer 2

Thanks for your suggestions, I attached my response.

Thanks
